# The Impact of Praise on Cooperative Behavior in Three-Player Public Goods Games and Its Gender Differences

**DOI:** 10.3390/bs14040264

**Published:** 2024-03-22

**Authors:** Jieyu Lv, Yingjun Zhang

**Affiliations:** 1Department of Psychology, School of Sociology and Psychology, Central University of Finance and Economics, Beijing 102206, China; 2Mental Health Education and Counseling Center, Beijing Normal University, Beijing 100875, China; yjzhang@bnu.edu.cn; 3Ideological and Political Work Team Training and Research Center, Beijing Normal University, Beijng 100875, China

**Keywords:** praise, three-player public goods game, gender differences, inferential social learning perspective

## Abstract

Previous research has primarily focused on static factors influencing cooperative behavior in social dilemmas, with less attention given to dynamic factors within group social interactions, such as positive feedback received during interactions, i.e., praise. This study, through a between-subjects online experiment with a single-factor, two-level design (praise: public praise/no praise), investigates the impact of praise on cooperative behavior changes across two rounds of a three-player public goods problem. Results revealed the following: (1) A positive correlation between individuals’ contributions across two rounds and a negative correlation with the number of correct answers in rule comprehension questions were evident; for men, a correlation between rule comprehension and first-round contributions was observed. (2) Multilevel model results showed that praise, role, and rule comprehension significantly positively affected contribution changes across two rounds; gender did not significantly affect contribution changes. Specifically, under public-praise conditions, contribution changes were greater. Publicly praised individuals showed positive or negative behavior changes, while those not praised in the same group showed positive changes. Men contributed significantly more in the first round than women, with no gender difference found in contribution changes. Rule comprehension positively predicted contribution changes, indicating that more correct answers led to greater positive changes in contributions. These results not only support the inferential social learning perspective, suggesting that through praise, individuals can infer external world perceptions and self-evaluations, affecting both the praised (positively or negatively) and positively influencing non-praised individuals in the same group, but also provide a theoretical basis and intervention strategies for team and organizational management in groups.

## 1. Introduction

A group of three has always been one of the simplest phenomena for a group. Cooperative behavior is a common prosocial behavior that occurs in situations where individual interests conflict with collective interests [1], where individuals choose to sacrifice personal benefits in favor of the collective [2]. The public goods dilemma is a widely used experimental paradigm to examine cooperative behavior in groups of three. Cooperative behavior is influenced by various factors, including static factors, such as individuals’ social value orientations [3] and empathy [4], and dynamic factors, such as behavioral or information feedback during interaction. Most previous research has focused on static factors [5] because they are relatively simple to study. However, this study focuses on a dynamic factor in interaction, namely, information feedback (referred to as praise in this study), and whether receiving evaluations of one’s behavior during interaction, especially positive evaluations, can lead to changes in individual behavior.

Positive evaluations during interactions, known as praise or commendation [6], are expressions in the emotional valence dimension of communication content within interpersonal interactions, serving as assessments of others’ behaviors to encourage, reward, and influence them [7]. Previous literature has categorized praise into person-focused praise and process-focused praise. Person-focused praise is a judgment based on the outcome of an individual’s behavior and capabilities, while process-focused praise is a judgment based on the effort level an individual puts into completing a task [8,9]. Studies have also confirmed that person-focused praise (e.g., intelligence, appearance) may weaken the recipient’s intrinsic motivation, causing them to attribute failure to personal traits when facing setbacks, leading to a reluctance to accept greater challenges and potential learned helplessness, thus exhibiting a non-adaptive, helpless response pattern. However, process-focused praise can enhance the recipient’s intrinsic motivation, leading them to attribute failures to the level of effort, thereby showing more effort in response [9,10].

This study adopts a person-focused praise: ”*You are a person who values collective interest and has a spirit of dedication*”. Yet, some argue that contextually appropriate praise can more effectively improve individual behavior. Previous research on the impact of praise on individual behavior has mainly focused on variables such as motivation [11,12] and self-evaluation [10,13], with a less direct study on the role of praise in resolving conflicts between individual and mutual interests. Despite the scarcity of research, praise may be an important and effective means of resolving conflict problems. Praise is usually of no economic cost, and people enjoy being praised with little social penalty toward the praiser. To date, only Wang et al. (2018) [14] found that praise symbolized by gold medals could compensate for the loss of benefits caused by following social norms, thereby improving prosocial decisions in a variant of the ultimatum game. Studying the role of praise in resolving conflicts between individual interests and collective interests can theoretically expand relevant theories and resolve conflicts practically, thus holding significant theoretical and practical value.

### 1.1. Effects of Praise on Cooperation: The Inferential Social Learning Perspectives

The promotion of cooperative behavior through praise can be understood from the Inferential Social Learning Perspectives (ISLP), which posits that praise conveys evaluations about an individual by others [15]. Through these evaluations, individuals can infer information about the external world and their own selves, thus guiding their behavior in various contexts [15,16,17]. Praise serves as an effective means for self-understanding, suggesting that when individuals are praised by others or the members of their groups, they infer their behaviors align with social group standards. This leads to attempts to please others and actively manage one’s reputation, adjusting behavior to gain further praise and improve reputation within the group. Therefore, based on ISLP, we propose:

**Hypothesis** **1.***Praise promotes changes in individual cooperative behavior*.

### 1.2. Who Is More Influenced by Praise Feedback in a Group, the Praised or the Unpraised?

Is praise effective only for those who are directly praised, or does it impact all members of a group? This question has not been explored in previous research. Our study design allows us to address this effectively. The hypothesis suggests that both individuals who are praised and those who observe praise within a group become more cooperative. This is because praise not only reinforces the behavior of the praised individuals, making them more likely to continue behaving positively, but also influences observers in the group who, seeing others praised, may seek to improve their behavior for future praise. Therefore, we propose:

**Hypothesis** **2.***In the public-praise condition, both praised individuals and non-praised observers will increase their cooperative behavior, leading to positive changes in contributions from all three players in the public praise condition*.

### 1.3. Gender Differences in the Impact of Praise on Cooperative Behavior

While some studies, e.g., [18], suggest unique cooperative links among women, most research indicates that men are generally more cooperative. This may be rooted in evolutionary psychology, with men historically engaged in hunting, requiring more cooperation, compared to women’s gathering. However, in the context of receiving praise within a group, it is questioned whether men or women are more likely to change their behavior. Women, being more socially oriented and concerned with others’ opinions, may be more susceptible to behavioral changes following praise. Therefore, we propose:

**Hypothesis** **3.***While men may inherently display more cooperative behavior, women are more influenced by praise, leading to a greater increase in cooperative behavior among women when praised.*.

### 1.4. Innovations and Significance of This Study

This study explores the effects of praise on cooperative behavior in a three-player public goods problem, contributing in several ways: (1) It reaffirms the role of praise in promoting cooperative behavior in social dilemmas. (2) It introduces a novel investigation into how non-praised members within a group are influenced by praise. (3) It examines gender differences in response to praise, suggesting that women are more affected by praise than men. This study, conducted on the oTree platform [19], offers low-cost intervention strategies for social dilemmas, providing diverse solutions for social governance and community management. All materials, data, and analysis scripts are openly available (https://osf.io/ar8b6/ assessed on 19 March 2024), aligning with open science principles.

## 2. Materials and Methods

### 2.1. Participants

The sample size was predetermined using G*Power 3.1 software [20] based on a medium effect size (f=0.5) and significance level α=0.05, requiring 128 participants to achieve 80%(1−β) statistical power. Through online recruitment advertised by fliers and WeChat posts, 306 participants were recruited (including 148 males and 158 females, $M_{\mathrm{age}}=19.38$, $SD_{\mathrm{age}}=1.96$). After processing missing data, 6 missing data points were found (specifically in the first round of contributions), and since this data was for a dependent variable, the method used was to delete the missing data, leaving 300 data points. The remaining results for further data analysis were N=300, $M_{\mathrm{age}}=19.39$, and $SD_{\mathrm{age}}=1.98$. Participants received monetary compensation, which was converted into participant fees at a certain ratio. Payments were made via Alipay. Participants were compensated based on the final amount of tokens from the two rounds of the public goods game, with a conversion rate of CNY 0.5 per token. The compensation for participating in the experiment ranged from CNY 6.6 to CNY 23.3 (approximately equivalent to USD 0.92 to USD 3.26) for each participant.

### 2.2. Design and Materials

A single-factor (praise: public-praise condition, no-praise condition) between-subjects experimental design was used. The public-praise condition refers to the feedback phase in the public goods dilemma, where Player 1 in the group is openly praised: “*This round of investment has ended, and these amounts have been fully deposited into your public account. From this round of investment, it can be seen that you are a person who values collective interest and has a spirit of dedication*”. In contrast, the control condition (no-praise condition) is presented with the investment results during the feedback phase as “*This round of investment has ended, and these amounts have been fully deposited into your public account*”. The dependent variable is cooperative behavior in the public goods dilemma before and after information feedback, namely, the number of tokens donated by participants to the public account in the first and second rounds in a real situation set up through the Internet.

#### 2.2.1. Two Rounds of Three-Player Public Goods Dilemma

This task involves multiple rounds of investment, with each team member having their own personal account, and in each round, every team member receives 100 tokens. Each group has a public account, and each member can invest a certain number of tokens $X_{i}$ into the public account as they wish ($0\leq X_{i}\leq 100$). The tokens invested in the public account are doubled and then evenly distributed to each team member’s personal account. The final amount in each person’s personal account is $\left(100-X_{i}\right)+\left(X_{i}+Y\right)\times 2/3$.

#### 2.2.2. Rule Comprehension Task

After the instructions are presented, three questions are displayed for a rule comprehension task to check if participants have accurately understood the task instructions. The specific questions are as follows: *“1. If you invest 5 tokens, and the other two people invest 55 and 60 tokens, respectively, how many tokens will you receive from the public account? 2. If you invest 20 tokens, and the other two people invest 55 and 60 tokens, respectively, how many tokens will you receive from the public account? 3. If you invest 44 tokens, and the other two people invest 55 and 60 tokens, respectively, how many tokens will you receive from the public account?”* There are three options for each question from which the participants select. Participants have to choose a correct answer from three options.

### 2.3. Procedure

The experiment was conducted online using the oTree platform, which allows multiple participants to participate in interactive experiments online despite geographical distances. We recruited participants in batches of six, and each group of three participants joined a virtual room, with every participant receiving a unique online link. Within these virtual rooms, participants could interact and exchange data. They were randomly divided into two groups, each consisting of a three-player public goods dilemma group. After grouping, the experimenter privately sent each student their room number and label number; e.g., “Room1, s001” indicates room number 1, label number s001. This setup ensured that each participant’s link was unique, allowing them to log into the corresponding experimental interface based on their room and label number. This study was conducted online, and room and label numbers were distributed randomly. Participants were unaware of their interaction partners, ensuring anonymity. Participants entered the webpage and filled in personal information, and the official experiment began. The whole experiment lasted approximately 15 min.

The specific experimental interface first presented a personal information collection page, including the lowercase initials of the name, age, and gender (for additional details regarding the procedure, please refer to the Appendix A). Next, participants were presented with the experimental instructions, followed by the game rules explanation for two rounds of the three-player public goods dilemma. If participants understood these tasks, they would proceed to answer three rule comprehension task questions. After completing the rule comprehension task questions, participants were informed of their role allocation, such as “*Your role: You are Player 1*”. Then, the first round of investment decisions began. Participants had to decide how much of the 100 tokens they could dispose of to invest in the public account. After participants invested, the program provided them with information feedback. In the feedback phase, there were two possibilities: one was the public-praise condition, that is, to give positive praise to Player 1 in a group, while the other two members (Player 2 and Player 3) would not receive praise. The other possibility was the no-praise condition, in which all three members of the group (Player 1, Player 2, and Player 3) would receive no praise feedback. After the feedback, participants made their second round of investment decisions. After the decision, the program tallied the total number of tokens participants received in the two rounds of the public goods dilemma. Finally, based on the total number of tokens received by participants, the experimental compensation was paid according to the number of total tokens. Participants left their Alipay information.

## 3. Results

We first present descriptive statistical information, showing the correlation levels of the main core variables, followed by the establishment of a multilevel model. This model incorporates factors affecting the change in the dependent variable, the contribution over two rounds, ultimately including praise, gender, the number of correct answers in rule questions, and role as fixed effects; with the group as a random effect, the multilevel model was analyzed with the change in contribution over two rounds as the dependent variable.

### 3.1. Statistical Information on the First- and Second-Round Contribution Values and the Change in Contribution Values under Different Praise Conditions

Table 1 shows the means, standard deviations, and 95%CIs for the first-round contributions, the second-round contributions, and the change in contributions under the conditions of public praise and no praise.

### 3.2. Correlation Coefficients and Gender Differences in Correlations

#### 3.2.1. Overall Results of Correlation Analysis

The results of the correlation analysis are presented in Figure 1, where the overall trends are shown in Figure 1a, indicating a positive correlation between the first-round contribution and the second-round contribution (r=0.67), and a negative correlation with the change in contributions (r=−0.46) and a certain negative correlation with the number of correct answers in rule questions (r=−0.33).

#### 3.2.2. Gender Differences in Correlation Analysis

Further subdividing into male and female participants, as shown in Figure 1b–d, reveals differences in the correlation between the first-round contribution and the number of correctly answered rule questions among genders. Specifically, for female participants, the first-round contribution positively correlates with the second-round contribution (r=0.49) but negatively with the change in contributions (r=−0.39). For male participants, besides a positive correlation between the first and second-round contributions (r=0.67) and a negative correlation with the change in contributions (r=−0.50), a negative correlation was found between the number of correctly answered rule questions and the first-round contribution (r=−0.59) and a positive correlation with the change in contributions (r=0.46). This indicates that for male participants, their cooperative behavior in the first round is inversely proportional to their accuracy in rule questions, suggesting those who answered incorrectly tended to cooperate more in the three-player public goods dilemma. Moreover, those males who answered the rule questions correctly showed a positive correlation with the change in their cooperative behavior.

### 3.3. Multilevel Model: Exploring the Effects of Various Factors on the Change in Contributions over Two Rounds

The multilevel model analysis, with individual-level variables (praise condition, gender, the number of correct answers in rule questions, and role) as fixed effects and group-level variables as random effects, examines the impact on the change in contributions over two rounds. The model includes a random intercept, allowing for variability in baseline changes in contributions across groups.

The mathematical formula for the model is
(1)$$Y_{ij}=\beta_{0}+\beta_{1}X_{1ij}+\beta_{2}X_{2ij}+\beta_{3}X_{3ij}+\beta_{4}X_{4ij}+u_{0j}+\epsilon_{ij}$$
where

$Y_{ij}$ represents the dependent variable, the change in contributions of the *i*th individual in the *j*th group.$X_{1ij}$ to $X_{4ij}$ represent the independent variables for each individual, including praise condition, role, gender, and the number of correct answers in rule questions, respectively.$\beta_{0}$ to $\beta_{4}$ are the fixed effect coefficients indicating the average impact of each independent variable on the dependent variable.$u_{0j}$ is the random intercept for the *j*th group, allowing for baseline variability in changes in contributions across groups.$\epsilon_{ij}$ is the error term, representing random variation at the individual level not explained by the model.

The results indicate significant positive effects of public praise, the number of correct answers in rule questions, and the role on the change in contributions over two rounds. Specifically, the praise condition significantly influenced the change in contributions, with public praise leading to a higher average increase compared to no praise. Gender showed a negative impact, though not significant, suggesting a possible smaller change in contributions for males compared to females. The number of correct answers in rule questions had a significant positive effect, indicating that a higher accuracy in understanding the rules led to a greater change in contribution behavior.

#### 3.3.1. The Impact of Praise on Individual Contribution Behavior

The model results showed a significant positive impact of the praise condition on the change in contributions between the two rounds, with $\beta_{1}=7.64$, SE=3.35, and t=2.28. As illustrated in Table 1 and Figure 2b, the change in contributions under the public-praise condition was significantly higher than that in the no-praise condition. Although there was no difference in the first-round contributions (t(298)=0.003,p=0.9968,Cohen’sd=0.000049), a significant difference was observed in the change in contributions (t(298)=−3.426,p=0.0007,Cohen’sd=−0.424), indicating a greater change in contributions in the public praise group.

#### 3.3.2. Differences in Contribution Behavior between Praised and Non-Praised Members within Groups

In the groups, the individual assigned to role 1 was the target of public praise. The model indicated a significant positive effect of role on the change in contributions between the two rounds, with $\beta_{2}=2.92$, SE=1.10, and t=2.66. A Kruskal–Wallis test comparing the change in contributions among players (1, 2, and 3) within the public-praise condition revealed significant differences (p=0.00022), as shown in Figure 2c. Specifically, Player 1, who received public praise, exhibited a significantly different change in contributions compared to Players 2 and 3, with no significant difference between Players 2 and 3. This suggests that being publicly praised (Player 1) results in a larger variability in contribution behavior, potentially leading to both increases and decreases, whereas the behavior of non-praised members (Players 2 and 3) tends toward positive changes.

#### 3.3.3. Gender Differences in Contribution Behavior and Changes in Contributions

Figure 2e shows that males (blue) had higher contributions in both the first and second rounds compared to females (pink), but there was no significant difference in the change in contributions between genders. An independent samples *t*-test revealed significant gender differences in the first-round contributions (t(298)=4.69,p<0.001,Cohen’sd=0.54) and the second round (t(298)=3.75,p<0.001,Cohen’sd=0.43) but not in the change in contributions (t(298)=1.30,p=0.194,Cohen’sd=0.15), indicating no detectable gender differences in the change in contributions.

Figure 2f illustrates different gender dynamics under public-praise and no-praise conditions. In the no-praise condition, there were no gender differences in either the first- (p=0.987) or second-round contributions (p=0.602). However, under the public-praise condition, gender differences were significant in both rounds, with a notable increase in the change in contributions for females compared to males under public praise (p=0.0058), suggesting that males displayed higher initial cooperative behavior, possibly due to awareness of being observed and evaluated.

#### 3.3.4. The Effect of Rule Comprehension Task Accuracy on Individual Contribution Behavior

Participants were divided into four groups based on the number of correctly answered rule comprehension questions: zero correct (n=36), one correct (n=48), two correct (n=20), and three correct (n=196). The model further demonstrated a significant positive prediction of the number of correctly answered rule comprehension questions on the change in contributions between the two rounds, with $\beta_{3}=3.94$, SE=0.91, and t=4.34 (see Figure 2g). Significant differences were observed in the first-round contributions among the four groups ($F\left(3,296\right)=14.64,p<0.001,\eta_{p}^{2}=0.148$), with the group answering one question correctly contributing the most on average. The analysis of the second-round contributions and the change in contributions also revealed significant differences, underscoring the complex relationship between rule comprehension accuracy and cooperative behavior in the public goods game.

## 4. Discussion

Through a simple experimental design, this study investigates the impact of public praise versus no praise on cooperative behavior and its change in a three-player public goods game. The findings revealed positive correlations between individuals’ contributions across rounds and a negative correlation with changes in contributions and the number of correct answers to rule comprehension questions. Moreover, multilevel models revealed that public praise significantly positively influences changes in contribution behavior, with gender showing no significant impact on changes but significantly affecting initial contributions. Men tended to be more cooperative initially, but behavior changes due to influence did not differ by gender. Correct answers to rule questions positively impacted contribution changes.

### 4.1. The Impact of Praise on Cooperative Behavior or Changes in Cooperative Behavior

#### 4.1.1. Overall Impact

The results of this study support Hypothesis 1, indicating that praise promotes cooperative behavior in a three-player public goods problem. The multilevel model results show a significant positive effect of praise on changes in contributions across two rounds, representing how individuals adjust their behavior after receiving feedback. Positive changes in contributions suggest increased cooperation post-feedback, whereas negative changes imply reduced cooperation. Publicly praised groups showed significantly higher increases in cooperative contributions compared to non-praised groups, highlighting the effectiveness of public praise in enhancing cooperative behavior. This aligns with previous research [21], suggesting a positive correlation between praise and cooperation rates, explained through the lens of ISLP. Public praise activates reputation mechanisms, motivating individuals to please others and proactively manage their reputation. Although public praise has a promotional effect on changing cooperative behavior, the long-term effects of this action cannot be explored in the experimental design of this study.

#### 4.1.2. The Differential Impact of Receiving Praise and Seeing Others Receive Praise within the Public-Praise Condition on Cooperative Behavior

Partially supporting Hypothesis 2, our data did not back the idea that personal praise directly increases one’s cooperative behavior. However, they did support the notion that observing teammates being praised enhances one’s cooperation. This highlights the complex role of public praise: while directly praised individuals may not always increase their cooperation, their teammates often do, suggesting that public praise generally elevates group cooperation. This phenomenon can be interpreted through the lens of ISLP, suggesting that public praise signals an environment that encourages cooperation, thereby influencing all group members toward more cooperative behavior despite the uncertainty in the direct recipient’s response to praise. This underscores the nuanced impact of public praise in group settings, marking a key finding of our experiment.

#### 4.1.3. Do the Effects of Praise on Behavior Differ by Gender?

The results are affirmative, suggesting that men are inherently more cooperative than women. However, in environments where praise is given, women are more susceptible to situational influence, unlike men. Our multilevel model results show that there is no statistically significant difference in the impact of gender on the change in contributions across two rounds. Thus, there is no significant gender difference in the change in cooperative behavior. However, our additional analysis revealed that in the first round of contributions, men’s cooperative behavior was stronger than that of women, consistent with previous research and our initial hypothesis. At the same time, we found that there was no gender difference in the contributions in the first round under the no-praise condition; but under the condition of public praise, there was a gender difference in the first round of contributions, specifically, men’s contributions were higher than women’s. However, under the condition of public praise, the change in contributions across two rounds was higher for women than for men. This indicates that praise leads to an increase in cooperative behavior changes in women. This result aligns with our prediction and is consistent with previous research findings by van Vugt et al. (2017) [22], but it contradicts the conclusions of the meta-analysis by Spadaro et al. (2023) [23], which did not find a significant gender difference in cooperative behavior in social dilemmas. Our results imply that women are more sensitive to praise, more situational, and more likely to change their behavior, particularly in a positive direction. This is inconsistent with the findings of Lessard et al. (2015) [24], which did not observe a significant main effect of gender or an interaction between gender and type of praise. It aligns with the findings of Henderlong et al. (2007) [25]: after experiencing a certain failure, process-based praise enhanced the motivation of fourth- and fifth-grade girls but did not affect boys’ subsequent motivation.

### 4.2. The Relationship between the Understanding of Rules, Cooperative Behavior, and Behavioral Change

In our introduction, we did not hypothesize a relationship between understanding the rules and cooperative behavior. This was based on the expectation that most participants would be able to comprehend the rules. The initial idea was to exclude participants who did not answer all the rule comprehension questions correctly. However, our data showed that a large portion of participants did not fully understand the rules. Furthermore, the correlational analysis revealed a relationship between rule comprehension, the initial contributions, and the change in contributions across two rounds. Results from multilevel modeling indicated a positive relationship between rule comprehension and changes in contributions across the two rounds. That is, the number of correct answers in rule comprehension questions was associated with a greater likelihood of changes in cooperative behavior. However, this study can only highlight this relationship at a surface level and cannot identify whether there is an underlying third variable that covaries. It is possible that individuals who answered more rule comprehension questions correctly are more likely to have higher social adaptability, rule learning ability, or stronger inferential social learning skills, enabling them to adjust their behavior more effectively under clear rules. Simultaneously, our data found that, for male participants only, the number of correctly answered rule comprehension questions was negatively correlated with the initial round of contributions. This finding supports the perspective that people cooperate because they are confused about the situation or problem, and the safest approach appears to be to cooperate. Previous research by Andreoni (1995) [26] suggests that people cooperate not because they genuinely want to but because they do not understand the rules in public goods problems, leading them to exhibit cooperative behavior.

### 4.3. Limitations and Future Directions of This Study

Although this study has strengthened the understanding of the positive role of praise in cooperative behavior, suggesting praise as a powerful intervention measure in cooperative behaviors within social dilemmas, it also has its limitations.

Firstly, within the same group, why does praise have both positive and negative effects on the individuals who receive it? This study does not provide a clear explanation behind the effect of praise on the target of the praiser. The dual impact of praise within a group setting raises intriguing questions about the underlying psychological processes. The variance in response can stem from individual differences in self-esteem, perceived sincerity of the praise, and social comparison effects [13]. Individuals with high self-esteem may respond positively to praise, seeing it as reinforcement of their self-view, while those with lower self-esteem may question the praise’s authenticity, leading to negative effects. Additionally, the social context of praise—whether it fosters a competitive or a supportive atmosphere—can influence its impact. This complexity suggests a need for a nuanced understanding of how personal and situational factors interact to shape the response to praise. Further research can explore these dimensions to provide a clearer explanation of the mixed effects observed.

Secondly, this research focuses on the effectiveness of praise in cooperative behavior in social dilemmas without exploring the potential role of blame. Evaluations from others can be both positive and negative [27]. Praise and blame often need to be considered together [28]. Therefore, future research can examine the impact and psychological mechanisms of cooperative behavior in social dilemmas under the combined effect of praise and blame in situations where blaming is allowed.

Thirdly, although this study found a negative correlation between the number of correct answers to rule comprehension questions and the initial round of contributions and multilevel modeling suggested a positive predictive role of rule comprehension on the change in contributions across two rounds, it does not provide a deeper or more plausible psychological mechanism between the number of rule questions answered correctly and cooperative behavior, including changes in cooperative behavior.

## 5. Conclusions

This study investigated the impact of praise on cooperative behavior in a three-player public goods dilemma and its variation by gender through an experimental design contrasting conditions of public praise versus no praise. The findings underscore the multifaceted role of praise in enhancing cooperative behavior, showing that public praise significantly increases the likelihood of cooperative behavior changes across two rounds. These changes are influenced by factors such as the individual’s role in the group and their ability to understand the rules of the game, illustrating the complex interplay between social feedback and cooperative dynamics. Furthermore, the study highlights gender differences in response to praise, with public praise having a more pronounced positive effect on women’s cooperative behavior changes than on men’s. These results contribute to the understanding of ISLP, suggesting that positive social feedback not only affects the praised individuals—potentially in both positive and negative directions—but also positively influences other group members, thus offering insights into how praise can serve as an effective intervention strategy in social dilemmas.

## Figures and Tables

**Figure 1 behavsci-14-00264-f001:**
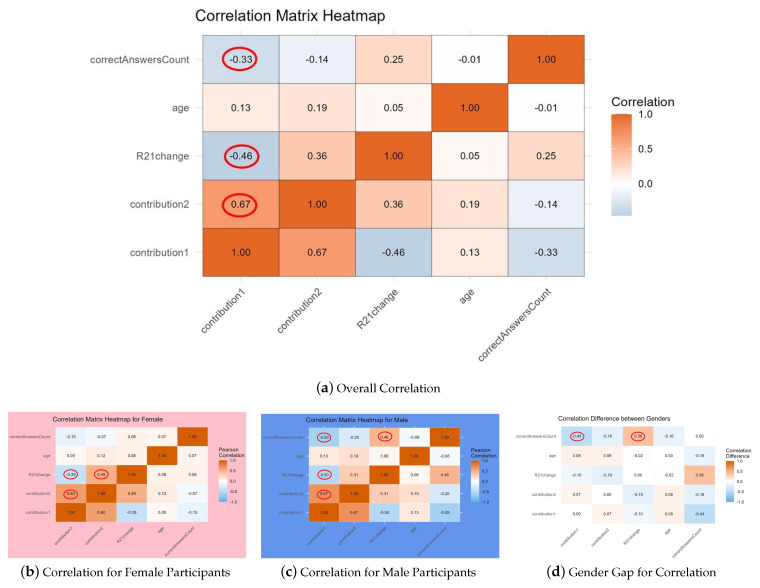
Heatmap of correlations among key variables. Red circles are used to indicate higher correlation coefficients.

**Figure 2 behavsci-14-00264-f002:**
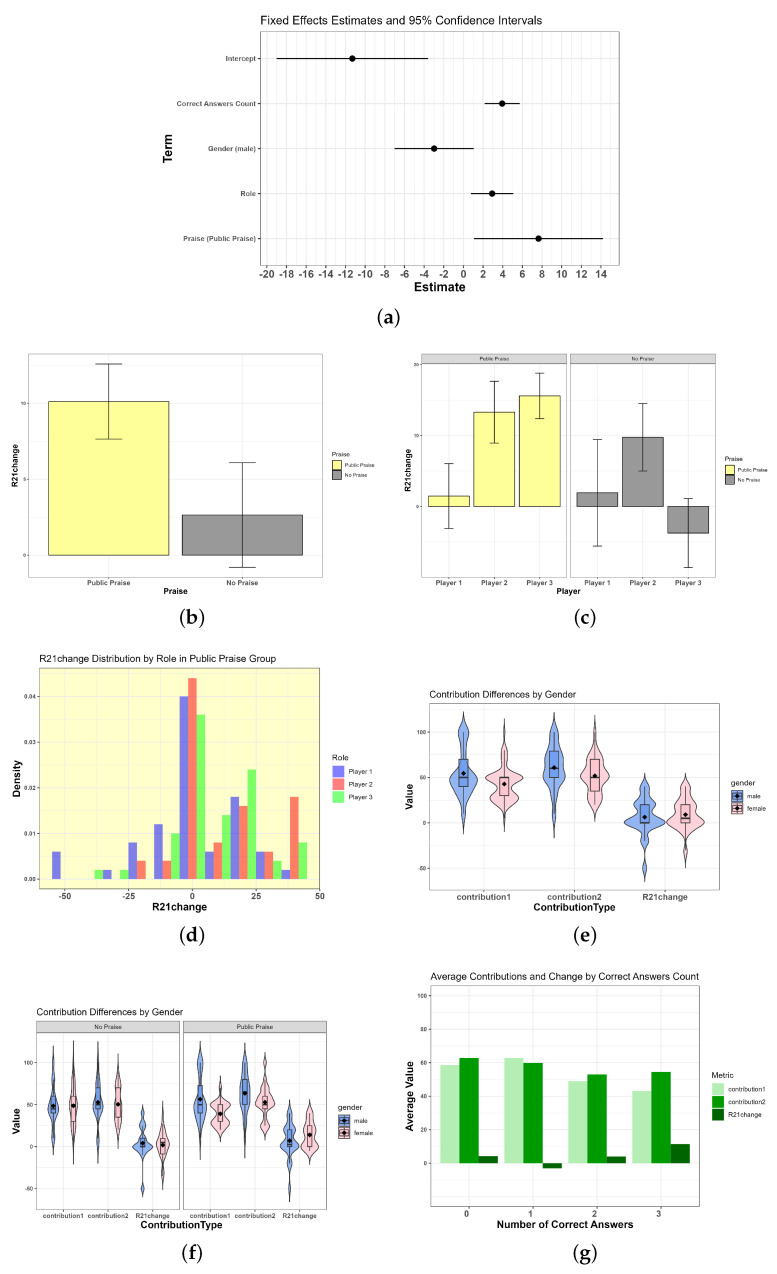
Fixed effects in the multilevel model analysis. (**a**) Fixed effects estimates and 95% confidence intervals; (**b**) praise effects on contribution; (**c**) behavioral difference based on role for being praised players and not being praised players in public-praise condition; (**d**) the contribution change based on role between Round 1 and Round 2 in public praise condition; (**e**) gender difference; (**f**) gender difference in various praise conditions; (**g**) rule understanding test.

**Table 1 behavsci-14-00264-t001:** Descriptive statistics of contributions in the first and second rounds and the change in contributions by role/gender.

			Contribution1		Contribution2		R21change	
	Role/Gender	*n*	*M(SD)*	95% CI	*M(SD)*	95% CI	*M(SD)*	95% CI
Public Praise	Overall	204	48.51 (22.57)	[45.41, 51.61]	58.63 (21.38)	[55.69, 61.56]	10.12 (17.91)	[7.66, 12.58]
No Praise	Overall	96	48.52 (22.22)	[44.08, 52.97]	48.16 (23.24)	[43.65, 52.67]	2.65 (16.99)	[−0.75, 6.04]
Public Praise	Player 1	68	54.41 (25.91)	[48.25, 60.57]	55.88 (25.17)	[49.90, 61.86]	1.47 (18.83)	[−3.00, 5.95]
Public Praise	Player 2	68	44.35 (24.08)	[38.63, 50.08]	57.65 (23.38)	[52.09, 63.20]	13.29 (18.02)	[9.01, 17.58]
Public Praise	Player 3	68	46.76 (15.35)	[43.12, 50.41]	62.35 (13.51)	[59.14, 65.56]	15.59 (13.26)	[12.44, 18.74]
No Praise	Player 1	32	47.75 (25.90)	[38.78, 56.72]	49.69 (22.58)	[41.86, 57.51]	1.94 (20.81)	[−5.27, 9.15]
No Praise	Player 2	32	49.69 (10.85)	[45.93, 53.45]	59.44 (17.17)	[53.49, 65.39]	9.75 (13.19)	[5.18, 14.32]
No Praise	Player 3	32	48.12 (26.87)	[38.81, 57.44]	44.38 (18.91)	[37.82, 50.93]	−3.75 (13.50)	[−8.43, 0.93]
Public Praise	Male	112	56.39 (25.57)	[51.66, 61.13]	63.39 (22.38)	[59.25, 67.54]	7.00 (19.61)	[3.37, 10.63]
Public Praise	Female	92	38.91 (13.01)	[36.26, 41.57]	52.83 (18.62)	[49.02, 56.63]	13.91 (14.82)	[10.88, 16.94]
No Praise	Male	34	48.47 (23.08)	[40.71, 56.23]	52.65 (23.14)	[44.87, 60.43]	4.18 (19.27)	[−2.30, 10.65]
No Praise	Female	62	48.55 (21.92)	[49.09, 54.01]	50.35 (18.98)	[45.63, 55.08]	1.81 (15.70)	[−2.10, 5.71]

## Data Availability

The data presented in this study are available on request from the corresponding author.

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
