# Peer review of "The Impact of Praise on Cooperative Behavior in Three-Player Public Goods Games and Its Gender Differences"

_behavsci, 2024, doi:10.3390/bs14040264_

Round 1

Reviewer 1 Report

Comments and Suggestions for Authors

The central theme of the work is interesting, however there is a lack of a theoretical framework of reference as well as a better description of the experiment.

There are important contributions to the literature that are not considered in the work, for example: Roemer, JE (October 1, 2013). Kantian optimization: an approach to cooperative behavior. Retrieved May 5, 2022, from the Cowles Foundation for Research in Economics: https://cowles.yale.edu/sites/default/files/files/pub/d18/d1854-r.pdf

Roemer, J.E. (2019). A theory of cooperation in games with application to market socialism. Journal of Social Economy, 77(1), 1-28.

There are entire sentences whose meaning I cannot understand, for example

"Groups of three, a firm tripod, the topic of discussion has always been one of the simplest phenomena in a group: the crowd emerges with three." (lines 24 and 25)

        or like the following:

"The public goods dilemma is a common game theory framework for discussing cooperative behavior in groups of three." (lines 30 and 31)

As I understand it, the central problem with public goods is the possibility of stowaways, an issue that is not considered at work and not necessarily in groups of 3.

People say that

``Positive evaluations during interactions, known as praise or flattery, are expressions in the emotional valence dimension of the content of communication within interpersonal interactions, and serve as an evaluation of the behaviors of others to encourage, reward and

influence them."

There is a wide range of experiences in this type of recognition. In former socialist countries, titles such as hero of socialist work or hero of the country, etc., were normally awarded in public. They do not appear to have had a major impact on the individual's attitude toward the group. The work does not make any reference to these types of experiences, which I believe are important when it comes to understanding the need for incentives for collective behavior.

The long-term effect of this type of recognition does not seem to be visible in a two-stage game, until when does public recognition prove to be a better incentive than monetary recognition to improve collective behavior? it is not clear. This point must be highlighted within the conclusions.

Finally, I understand that the overall experiment should be presented better to aid general understanding. It's unclear (at least to me) how each player's pro-collaboration behavior is made public.

In general I think it is necessary to improve the general presentation of the work. The statistical part can be left for the appendix.

The overall presentation of the experiment needs to be improved.

It should be clearer how awards for collective behavior are made public, but I understand that public recognition is the center of the discussion.

The conclusions should clarify the limitations of the experiment and the relationship between the short and long term of public incentives.

The references must be expanded, there is a wide range of literature referring to the subject, from the seminal works of Balga-Maskin, to the most current ones of Roemer. Considering the works of Coase or Barret's pardox does not seem to be a bad idea either.

Comments on the Quality of English Language

/

Reviewer 2 Report

Comments and Suggestions for Authors

See the pdf file.

Reviewer 3 Report

Comments and Suggestions for Authors

Referee Report on:

The Impact of Praise on Cooperative Behavior in Three-Player Public Goods Games and Its Gender Differences: From Inferential Social Learning Perspectives

The paper reports evidence from an online VCP experiment, where subjects received personal praises of process praises. As it is constructed, the paper is intersting: it is the first time that this type of experimental design has been applied to VCM models, or more generally, to cooperation.  

Also, the model is innovative; it is a three-players dynamic sequential VCM model.

However, I think the paper greatly needs to be revised before it reaches a publishing state.

Major points:

1.       The paper is writtern in a very unprofessional way. It is complicated to read it and it is hard to understand what the author wants to convey as the final and more important point (praising is a good thing when groups are formed by women?). Even in the introduction, the first paragraph describes the results of a power analysis conducted on the subjects’ sample. This analysis should be reported in the section of the experimental design, in a footnote. Mistakes of similar kind are present in almost all sections. For example,  it seems that the VCM game was a sequential model, but it is not very clear, since the theory behind  the experiment is not reported (for VCM games, it is very important to report the value of a, the coefficent that increases the value of the individuals’ contributions). Secondly, it is not clear what the subjects actually observed before entering their choices (every player’s per- period contribution?). Thirdly, it is not clear why only one player was praised (the first one to choose?). Fourth, it is unclear how many periods participants  played and why that number of periods was selected.

2.       English editing is very poor. Also, there are a lot af typos (for example, in the front page, the department is wrongly spelled).

3.       The title is too long. The author does not need to report all parts of the research.

4.       The empirical part is confusing: too many sections and it is hard to uderstand which resulti s really new and interesting. Also, The empirical section should have proper tables, where statistical  tests are reported.

5.       The majority of the innovative results regard the behaviour in the first stage, while in the following stages, behaviour seems to converge among all players. This might be  a problem, since first period results are gnerally influenced by random factors and low levels of learning and understanding. Furthermore, it is a well established result in the VCM literature that in repeated contexts first stage behaviour is significantly different from the other stages. The author shoul comment on that.

6.       The experiment is very badly reported: how much the subjects earned? How was the recruiting  carried out? How much time the experiment lasted? And so forth…

7.       The theory should have its own section, where the VCM game is clearly reported. As it is, I do not understand the characteristics of the model.

Despite all these shortcomings, I think this paper has interesting points. I would like to see a revised version. My suggestion, though, is that the author re-write the paper in a professional way.

Comments on the Quality of English Language

The quality of English is very low.

Round 2

Reviewer 1 Report

Comments and Suggestions for Authors

The new version substantially improves the previous one and all suggestions were considered